# Active Flow Control on a Square-Back Road Vehicle

**Juan José Cerutti \***, **Costantino Sardu**, **Gioacchino Cafiero and Gaetano Iuso**

Dipartimento di Ingegneria Meccanica e Aerospaziale , Politecnico di Torino, 10129 Turin, Italy; costantino.sardu@polito.it (C.S.); gioacchino.cafiero@polito.it (G.C.) ; gaetano.iuso@polito.it (G.I.)

\* Correspondence: juan.cerutti@polito.it ; Tel.: +39-366-5948-411

**Abstract:** An experimental investigation focused on the manipulation of the wake generated by a square back car model is presented. Four continuously-blowing rectangular slot jets were mounted on the rear face of a 1:10 commercial van model. Load cell measurements evidence drag reduction for different forcing configurations, reaching a maximum of 12% for lateral and bottom jets blowing. The spectral analysis of the pressure fluctuations evidence, for all forced cases, an energy attenuation with respect to the natural case, especially close to the shedding frequency. An energy budget highlighted the most efficient forcing configurations accounting for both the drag reduction and the power required to feed the blowing system. Two main configurations are considered: the maximum drag reduction and the best compromise, yielding 5% drag reduction and a convenient energy balance. Particle Image Velocimetry (pPIV) and stereoscopic PIV (sPIV) experiments were performed allowing the three-dimensional reconstruction of the wake in the three considered configurations. Consistently with static and fluctuating pressure measurements, sPIV results reveal a dramatic change in the wake structure when the jets blow in the maximum drag reduction configuration. Conversely, the best compromise configuration reveals a wake structure similar to the natural one.

**Keywords:** active flow control; drag reduction; stereoscopic PIV; bluff body wake

## 1. Introduction

The growing concern on climate problems and pollution is pushing all the international institutions to introduce targets for road vehicles' emissions and promoting guidelines for the design of the new generations. The aerodynamics of the vehicle, especially those with a bluff shape, plays an important role on the consumption, thus directly influencing the emission of pollutants. It has been estimated that a reduction in the aerodynamic drag of a vehicle of 10% leads to a 5% reduction in fuel consumption [1]. In addition, the improved aerodynamics will also benefit electric vehicles, significantly increasing their range.

A large body of research focused on the study of the Ahmed body. This model is a simplistic representation of a vehicle, with a curved forehead to avoid flow separation, a straight core with rectangular cross section and a flat back, which may or may not present a slant angle. The main flow features that characterize the near wake of such object can be summarized as follows: a separation region on the rear window, which may or may not be present depending on the slant angle, two recirculation bubbles near the rear base, and a pair of nearly streamwise counter rotating vortices (often referred to as C-pillar vortices). The interplay between these structures has a huge impact on the total aerodynamic drag of the vehicle. Furthermore, any attempt to control the flow should be aimed at tampering with these structures to obtain an effective drag reduction.

The control methodologies are usually categorized into passive or active. Those belonging to the first category do not require auxiliary power for their functioning and for their control [2–4].

Passive devices are implemented for a wide range of applications, such as heat transfer enhancement [5], combustion processes [6], and flow control, with particular emphasis on drag

reduction [7,8]. In the case of vehicle aerodynamics, solutions such as flaps [9], vortex generators [10], streamlining the body-shape, and local body-shape modifications [11–13] have achieved a great success. Nevertheless, the major limitation of these devices is their capability of working at the maximum efficiency for one only configuration. Moreover, they present several homologation problems, mainly associated to safety issues produced by the aerodynamics appendices and the modifications of the body shape. Lastly, the presence of appendices often also hinders the aesthetics of the vehicle.

Active flow control represents an interesting and appealing alternative. In fact, the possibility of tuning the control parameters to optimize the overall system efficiency in different configurations is particularly amenable for practical applications. Moreover, the actuators do not modify largely the body of the vehicle since they have reduced dimensions and can be easily fitted to existing vehicles.

For realistic applications of an active flow control strategy, the reliability and the efficiency of the technique play an important role. One of the most promising techniques is based on continuous jets, where a certain number of jets are generally located near the vehicle's rear face to control the shedding from the four trailing edges. Roumeas et al. (2009) [14] numerically investigated this problem using four blowing slots arranged along the base of a semi-infinite body. In this work, they investigated the sensitivity of the wake to the blowing angle and to the blowing intensity, and they stressed the effect that the control mechanism has on the toroidal vortex. They investigated five equally spaced blowing angles between 30° and 90°, directed towards the center of the wake, and found that 45° is the most effective configuration. For this angle, they obtained a maximum drag reduction of 28.9% with respect to the uncontrolled case. The efficiency of the control system was also calculated, which gave an indication of the potential energy savings accounting also for the energy necessary to feed the control mechanism. A similar experimental study was performed by Littlewood and Passmore (2012) [15] where a Windsor body with a square back base was considered. Their control strategy was based on continuous blowing through a slot placed near the top edge of the vehicle's base. A maximum drag reduction of 12% was achieved for the maximum considered blowing speed, without accounting for the system efficiency.

Steady jet blowing located near the four edges of a square back model rear face was also studied by Zhang et al. (2018) [16]. The model was an Ahmed body presenting a slant angle of 25° on the top edge. The simultaneous application of the four jets led to a dramatic change of the model's drag (approximately 30%), linked to a strong difference in the flow topology, involving both the extent of the recirculation bubble and the C-pillar vortex.

Other solutions focused on unsteady manipulations of the wake, to exploit the natural instabilities of the flow or locally manipulate the flow structure [17–20]. This is widely used in several fields, such as jets excitation [21], induced drag reduction [22], and wall turbulence control [23]. In the case of road vehicles, Barros et al. (2016) [24] investigated the implementation of pulsed jets along with Coanda effect. They pointed out that a high frequency actuation is necessary to achieve drag reductions, whilst a broadband actuation may even lead to drag increments. Furthermore, the effectiveness of the system is even higher when the Coanda effect is exploited, with a smaller entrainment rate in the near wake (even in the steady blowing case) and achieving drag reductions as large as 20%.

An extensive effort has also been devoted to the investigation of the flow instabilities arising in the near wake of a square back model. Grandemange et al. (2013) [25] carried out an experimental investigation of the wake bi-stability varying the ground clearance ($h/W$, being $h$ the distance between the floor and the underbody and $W$ the model's width) for two different values of the model's aspect ratio ($H/W$ = 0.74 and 1.34, where $H$ indicates the model's height). They showed that, for values of $h/W$ >0.12, the wake may exhibit a bi-stability in the lateral or in the vertical direction, depending on the value of the rear face's aspect ratio.

Volpe et al. (2015) [26], Bonnavion and Cadot (2018) [27], and Li et al. (2019) [28] extended these results introducing the effect of yaw and pitch angles. Their results show that, even with a small yaw angle, the onset of a bi-stable behavior in the vertical direction could be detected. More complex is

instead the dependency on the pitch angle, where the lateral symmetry breaking mode appears only for a narrow range of angles ($\approx \pm 0.75°$).

Brackston et al. (2016) [29] developed a stochastic model for the suppression of the symmetry breaking modes. Their results, based on the implementation of active flaps near the rear face of a square back model, showed that the mitigation of SBM may lead to drag reductions as large as 24%.

The motivation of this study arises from the general problem of reducing the drag of road vehicles and in particular of a very bluff body configuration (i.e., vans and square-back trucks). The fundamental point that we wish to address in this work relates to the forcing configuration leading to drag reductions and how the near wake structure reorganizes in such control configuration. Furthermore, we are interested in defining a metric that gives a direct grasp of the actual profit gained controlling the near wake, hence obtaining an optimal forcing configuration.

This study presents information about bodies with aspect ratio $H/W > 1$ (Grandemange et al. (2013)) that have received less attention in the literature, despite their significance especially for commercial road vehicles. In addition to this, to the authors' best knowledge, there are few experimental data available that present the full reconstruction of the near wake of a commercial road vehicle in the natural and controlled cases.

We aimed at investigating an efficient active flow control technique based on continuous jets applied to a square-back van model, using different combinations of the slot jet actuators covering the four edges of the rear face. We assessed the effectiveness of the flow control system for a potential realistic application, introducing a definition of the control efficiency which accounts also for the losses in the pneumatic lines. This latter aspect has been often overlooked in previous investigations. To this end, the energy budget analysis was performed in detail and a multi-objective approach was introduced to identify the optimal configurations of the system evidencing the most profitable blowing conditions.

## 2. Materials and Methods

### 2.1. Wind Tunnel and Model

The experiments were carried out in an open circuit wind tunnel equipped with two fans located upstream of the test section. A rapid diffuser with three mesh screens inside connects the exit of the fan to the stagnation settling chamber. A convergent duct delivers the air to the test section, having a rectangular cross section of 1.2 m times 0.9 m of height and width, respectively, and total length 6.5 m. Furthermore, the test section was designed with a 1% divergence angle to account for the growth of the boundary layer on the walls.

The model, shown in Figure 1a, is a square-back road vehicle characterized by a scale factor of 1:10. This results in a model cross section of height $H = 0.2$ m and width $W = 0.17$ m. The rear face of the model presents a slant angle towards the center of the wake of 10° extending along its entire perimeter. This choice was motivated by the results of Barros et al. [24], who showed that the concurrent application of active flow control and Coanda effect is extremely beneficial for the drag reduction purposes. The model's ground clearance $h$ measured with respect to its bottom surface is equal to 20 mm, corresponding to a normalized value $h/H = 0.1$.

Two different approaches can be employed for wind tunnel testing of road vehicles: a first one that relies on a moving floor and a second one where the floor is stationary. In the latter case, it is fundamental to ensure that the boundary layer thickness growing on the floor is small compared to the clearance $h$. This is usually achieved by either mounting the model on a plate, where the boundary layer origin is fixed, or by providing a suction of the boundary layer. Stationary floor has been employed in several cases in the past (e.g., [8,16,25,30,31]).

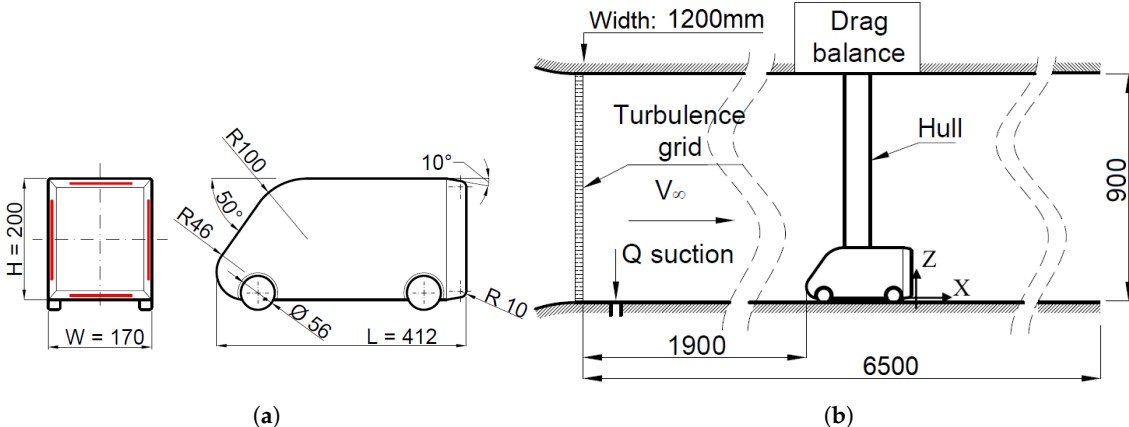

**Figure 1.** (**a**) Schematic representation of the model; the red continuous lines are representative of the injection slots of the active flow control system. (**b**) Schematic representation of the wind tunnel test section and indication of the reference frame. Dimensions are given in mm.

The mounting solutions can be differentiated as follows: a first solution where the model is connected to the floor through pins that very often correspond to the wheels and a second where a holding strut is located above the model.

In the present set of experiments, given the absence of a moving floor and the need to carry a significant amount of pneumatic lines and connections to the model (as described below, both used for measurement and for carrying the pressurized air to the jets), we opted for a configuration where the model was connected to a load cell mounted on top of the wind tunnel using an aerodynamically shaped holding hollow strut. To reduce to a minimum the aerodynamic interference of the strut, it is embedded within a hollow hull with a symmetric airfoil section and a relative thickness $t/c = 0.2$ and $c/W = 0.07$. In this way, the underbody flow resulted clean and unperturbed, and the hollow hull served the purpose of carrying the pneumatic lines to the pressure taps as well as to the blowing jets.

A sketch of the model and its positioning within the wind tunnel are presented in Figure 1b. The model is positioned 1900 mm downstream the inlet section through an instrumented strut attached to the model roof and directly connected to the balance.

The model frontal area presents a 3.1% blockage relative to the wind tunnel cross section, which increases up to 4.5% accounting also for the strut.

At the inlet of the test section, a grid having a meshlength equal to 65 mm and grid bars having thickness 20 mm was used to set the value of the incoming flow turbulence intensity (approximately 4%).

Furthermore, a suction slot of 250 mm × 6 mm is located about $1L$ ahead of the model (see Figure 1b) to reduce to a minimum the boundary layer growth on the wind tunnel floor. A vacuum pump provides the suction flow rate corresponding to a ratio $V_{suction}/V_{\infty} \approx 1$. We characterized the incoming boundary layer with dedicated hot wire measurements carried out $0.5L$ ahead of the model. The resulting ratio of the displacement thickness over the model's ground clearance is $\delta^*/h = 0.07$, in good agreement with values reported in other automotive investigations [32,33], where the rolling road was not taken into account due to limitations of the wind tunnel facility.

The freestream speed for all the tests was set to $V_{\infty} = 9$ m/s corresponding to a Reynolds number based on the model length of $Re_L = 2.5 \times 10^5$. The corresponding measured drag coefficient, $C_D = 0.465 \pm 5.66 \times 10^{-3}$, is in good agreement with values found in other investigations on similar road commercial vehicles (e.g., [8,34]).

*2.2. Flow Control Actuation*

Four independent actuators are located along the periphery of the model rear face, as shown in Figure 2a (red lines). The single actuator consists of a cylinder with a rectangular slot realized along

its length whose width is $h_j = 1$ mm ($\approx 0.006W$). The upper and lower slots are 104 mm in length ($\approx 0.61W$), while the lateral slots are 132 mm ($\approx 0.78W$). The circular section allows the actuator to change the jet orientation $\theta_j$, as shown in Figure 2a.

The jets feeding system is reported in Figure 2b. The Valves (V) and the flow meters (FM) were used for the mass flow rate regulation. In particular, the manometers (P) were installed along the pneumatic lines to accurately regulate the jets inlet pressure. These measurements are fundamental for the pressure losses evaluation necessary for the energy budget estimation.

The jet orientation $\theta_j$ was maintained constant and equal to 65° for all the forced configurations while the jet velocity was varied. The blowing speed for the upper and lower jets were independently controlled by the valves (V1–V2), the manometers (P1–P2), and two flow meters (FM1–FM2), as displayed in Figure 2b. The two lateral jets slot were instead simultaneously controlled through V4–FM3–P5 devices.

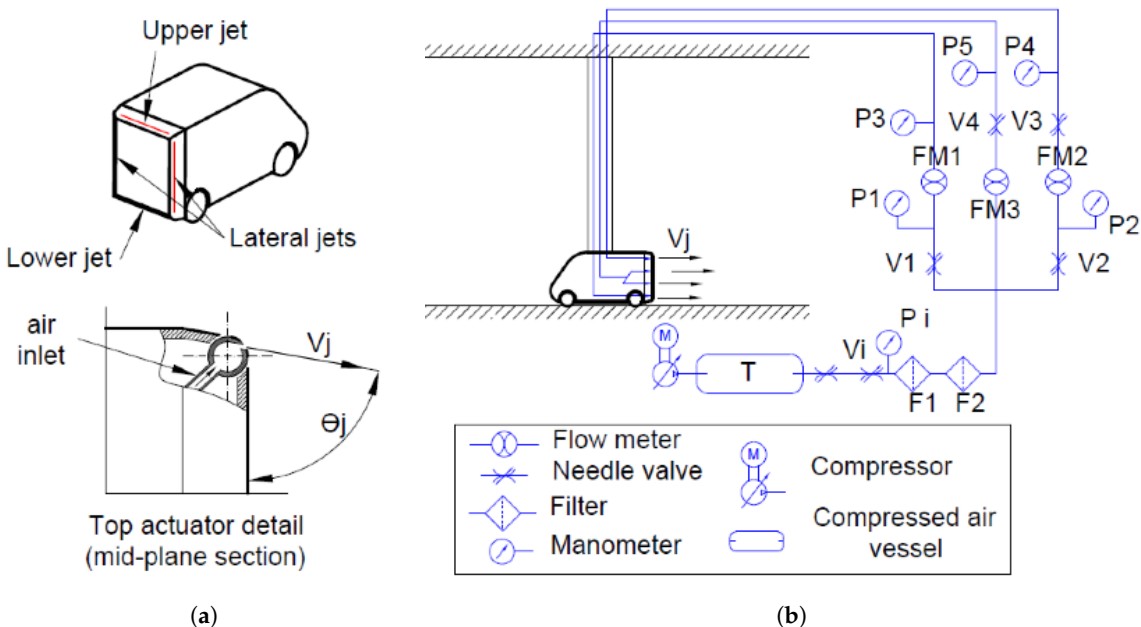

**Figure 2.** Flow actuation: (**a**) details of the slots; and (**b**) blowing speed control chain.

The four rectangular jets were initially tested to ensure the uniformity of the velocity profile. The measurements were carried out in wind-off conditions (i.e., at $V_\infty = 0$) and revealed that, at 1 $h_j$ from the jet exit section, all four jets have a uniform velocity profile for at least 95% of the slit length.

### 2.3. Measurement Techniques

The model is instrumented with two different types of pressure transducers: a multi-input pressure transducer for the mean static pressure measurements and microphones for fluctuating pressure measurements.

The mean pressure measurements were performed with a Scanivalve Smart ZOC 33 pressure transducer, capable of sampling 64 pressure channels simultaneously at a sampling frequency $f_s = 500$ Hz. The transducer has a full scale of $\pm 2.5$ kPa and accuracy equal to 0.15% FS. Pressure·inputs are connected to pressure taps distributed on the top, lateral, and rear planes of the model using plastic tubes with internal diameter $\phi = 1$ mm. The model rear face is instrumented with 31 pressure taps, while the remaining 33 are located along the other models surfaces (top, lateral, front). The mean length of the tubes is about 150 mm corresponding to a first resonance peak centered at 212 Hz.

The fluctuating pressure was measured using 16 electret capacitive capsules characterized by 9.8 mm external diameter and a sensing hole having diameter equal to 1 mm. Twelve microphones

were mounted on the rear face while four were positioned on the four surfaces adjacent to the rear face at a distance from the trailing edges equal to 10 mm. The microphones are characterized by a flat response in the range of frequency 0.005–13 kHz and a sensitivity of $-60 \pm 3$ dB. All sensors were first simultaneously calibrated using a Bruel & Kjaer probe, as reported in Sardu et al. (2016) [35]. A pinhole mounting configuration was adopted for both the model measurement and the calibration. The raw electric signal from the microphones is filtered to eliminate spurious contributions, as recommended by Sardu et al. (2017) [36].

The vehicle drag was measured using a Dacell UU-K002 load cell (full scale of $\pm 2$ $Kg_f$ and accuracy equal to 0.002%FS). The load cell signal is sampled using a NI-cDAQ chassis with a dedicated NI-9215 A/D converter module. The electric signal of the load cell is converted to drag through a calibration mapping. A repeatability campaign of measurements was conducted to mitigate the occurrence of outliers in the measurements. This allowed us to reduce the error down to a value of 0.20%.

The flow field structure in the near wake was investigated through two different setup: a first one in planes parallel to the model's rear face through stereoscopic PIV measurements (sPIV). Two Andor sCMOS 5.5 pixel cameras are located outside the wind tunnel with an angle between the line of sights of about $60°$. Both cameras are equipped with Nikon 60 mm lenses, and operated at a value of the aperture equal to $f_{\#}$ = 16. The field of view of each camera is $1.3W \times 1.1W$ in the $YZ$ plane. The proper focusing of the target plane is ensured by using in-house made Scheimpflug adapters. The illumination is provided using a Litron Laser Dual-Power 200 mJ/pulse operated in dual pulse mode at 15 Hz. The laser beam is shaped into a laser sheet using a spherical (1000 mm focal length) and a cylindrical ($-50$ mm focal length) lenses. The laser thickness in the region of interest for the measurements is about 1 mm. A schematic representation of the sPIV system is reported in Figure 3a. Both the cameras and the laser are located on parallel linear stages, allowing for their simultaneous streamwise displacement. We performed sPIV measurements in seven different planes in the streamwise direction, ranging between 0.82 $W$ and 1.35 $W$.

The flow was seeded using Fog Fluid 'Extra Clean', for PIV applications. The liquid was vaporized into small droplets of typical diameter of 1 μm, thus resulting in a Stokes number $\ll 1$.

In the sPIV case, the arrangement of camera and laser on parallel linear stages allowed us to perform the optical calibration at the beginning of the first test and at the end of the last one. We corrected for any small misalignment occurring during the system displacement performing the self-calibration procedure [37]. The final resulting error on the calibration was of the order of 0.05 pixels for every investigated case.

For each location and configuration investigated with sPIV, we acquired 2000 double-exposure statistically-uncorrelated images. The time delay between the two exposures was set to 60 μs, resulting in an average particle displacement of 6–8 pixels in the cross-wise plane. This allowed us to properly sample the flow within the $YZ$ plane, without incurring in the problem of loss of correlation due to particles leaving the illuminated plane.

The images of tracing particles were preprocessed to remove the historical minimum, to attenuate the effect of laser reflections on the wind tunnel walls and on the model (see Figure 3d). Image deformation and velocity vector field interpolation were carried out using spline functions [38,39]. A Blackmann weighting window was used during the correlation process to tune the spatial resolution of the PIV process [40]. The final window size is $64 \times 64$ pixel (corresponding to 3.55 mm $\times$ 3.55 mm), with 82% overlap. Considering the subpixel accuracy of the PIV process, and comparing it to the typical value of the mean flow component in the present investigation, we could assess that the average error on the velocity components is much smaller than 1%.

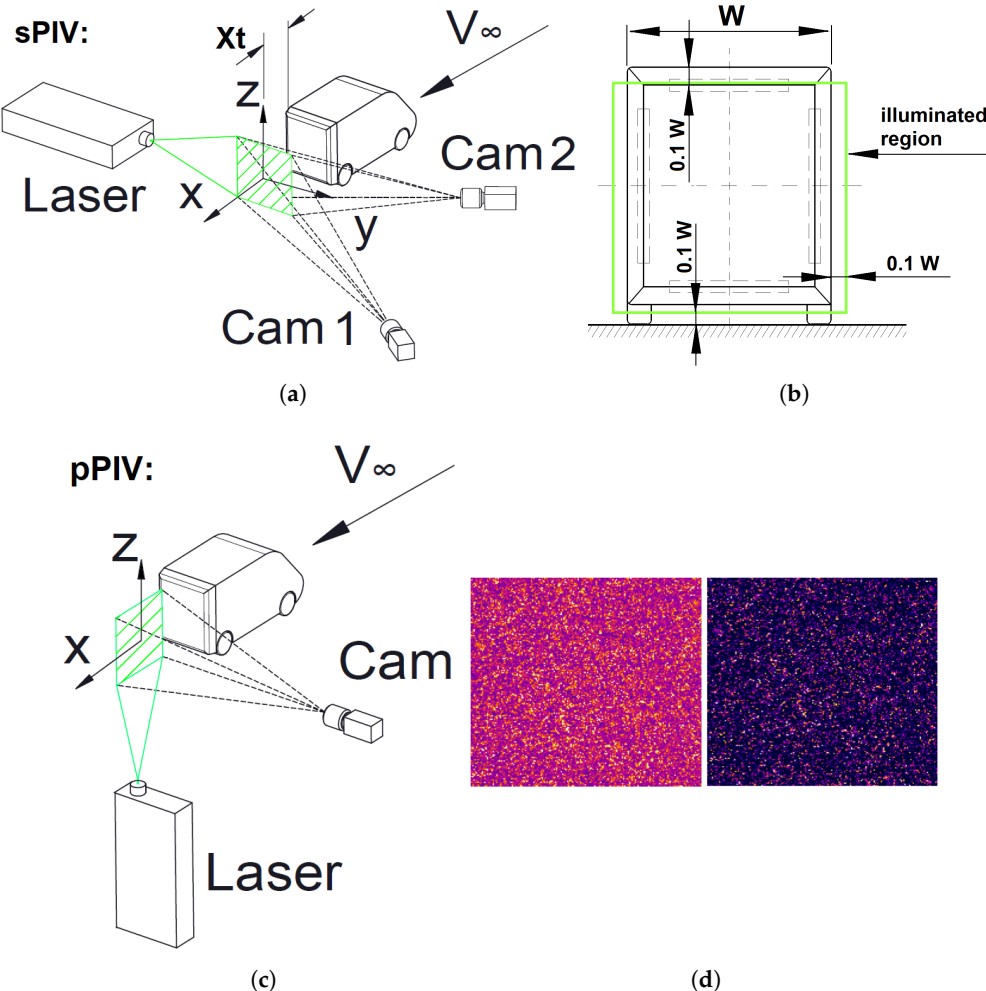

**Figure 3.** (**a**) Schematic representation of the sPIV system with indication of the imaged planes location. The investigated planes are $Xt/W = 0.82$, $0.91$, $1$, $1.09$, $1.18$, $1.26$, and $1.35$. (**b**) Detail of the illuminated region and spatial location with respect to the model's rear face. (**c**) Schematic representation of the pPIV system with indication of the imaged planes location. (**d**) Comparison between raw (left) and preprocessed (right) image after historic minimum removal.

Relevant information about the streamwise organization of the flow field was also gathered performing pPIV measurements in planes aligned to the symmetry plane of the model ($XZ$). The setup of the laser and camera is sketched in Figure 3c. The camera, equipped with a Nikon 105mm lens, images an area extending $1.1\,W \times 1.3\,W$ in the $X$ and $Z$ directions, respectively. We carried out measurements in two planes, $Y = 0$ and $Y = 0.2$ for each configuration, acquiring 3000 images for each case. The time delay between two exposures was set to 40 μs. The raw images were preprocessed in a similar fashion to the sPIV experiments. The final interrogation window size is $24 \times 24$ pixels (2.5 mm $\times$ 2.5 mm), with 75% overlap.

## 2.4. Statistical Convergence and Accuracy Analysis

A preliminary investigation was carried out to assess the appropriate total sampling time to guarantee the statistical convergence of each quantity. In fact, due to the wake unsteadiness, the life-time of the dominant structures and the random flow oscillations influencing the model base can vary significantly.

A statistical approach was used following the procedure of Bendat and Piersol (2010) [41] to determine the number of samples which granted a satisfactory value of the confidence interval.

In particular, it can be assumed that the distribution of the observations is normal with a mean value $\overline{X}$ and standard deviation $\sigma_X$. For a fixed value of the confidence interval $1-\alpha$, it is possible to obtain the Z-score ($Z(\alpha)$) associated with this confidence interval, which is tabulated for a given probability distribution. The margin of error is given by $\sigma_X \cdot Z(\alpha)/\sqrt{N}$, where $N$ indicates the number of observations. We then set the value of the number $N$ in order to get an error on the mean value $\overline{X}$ of $\pm\epsilon$. Table 1 summarizes estimated error, sampling time, and frequency for each one of the employed measurement techniques, assuming a confidence interval of 95%.

**Table 1.** Statistical convergence for a confidence level equal to 95% and given error on the mean values.

| Measured Quantity and Given Error | Sampling Time (s) | Sampling Frequency (Hz) |
|---|---|---|
| Load Cell ($\leq\pm0.05\%$) | 240 | 1000 |
| Static and dynamic pressure ($\leq\pm0.1\%$) | 240 | 40 |
| Pressure fluctuation ($\leq\pm1\%$) | 15 | 10,000 |

The uncertainty of each measured quantity was also estimated through the error propagation analysis in a chain of measurement considering the procedure introduced by Kline and McClintock (1953) [42] and Moffat (1988) [43]. As an example, the uncertainty on the $C_D$ involves three main parameters: the load cell output ($\overline{E_T} - \overline{E_0}$, where $\overline{E_T}$ and $\overline{E_0}$ are the time averaged output voltages of the load cell in wind on and off conditions, respectively), the load cell calibration constant $K_{lc}$, and the dynamic pressure of the undisturbed flow ($\overline{\Delta p}_{pit}$). The definition of the drag coefficient in terms of the previous parameters reads as

$$C_D = \frac{K_{lc}(\overline{E_T} - \overline{E_0})}{(\overline{\Delta p}_{pit})S} \tag{1}$$

and $S = H \cdot W$ is the cross sectional area of the model. The associated uncertainty to the drag coefficient is consequently evaluated by means of the error propagation relation accounting for the number of parameters $N_{par}$

$$\delta C_D = \sqrt{\sum_{i=1}^{N_{par}} \left(\frac{\partial C_D}{\partial x_i} \delta x_i\right)^2} \tag{2}$$

In Table 2, we report the uncertainties and the related percentage errors for drag coefficient, pressure coefficient, and total pressure coefficient in the wake.

**Table 2.** Evaluated uncertainties and errors.

| Measured Quantity | Uncertainty ($\delta$) | % Error |
|---|---|---|
| Drag coefficient | $\pm5.66\cdot10^{-3}$ | $\pm1.2$ |
| Pressure coefficient | $\pm1.83\cdot10^{-3}$ | $\pm0.9$ |

## 3. Results

The reference frame used in the present investigation is defined such that the $X$ direction is aligned to the mean flow, $Z$ is the vertical direction, and $Y$ completes the triad, as already presented in Figure 1. The origin of the reference frame is located on the wind tunnel floor in correspondence of the rear face of the model, along its symmetry plane. The mean velocity components are indicated as $\overline{V}_X$, $\overline{V}_Y$, and $\overline{V}_Z$, while the velocity fluctuations are $v_X$, $v_Y$, and $v_Z$, such that $V_i = \overline{V}_i + v_i$. We first focus on the description of the natural flow, i.e., without forcing; subsequently, the forced configurations are presented and discussed.

### 3.1. Natural Flow

In Figure 4a, we report the mean pressure coefficient on the model side and in the *XZ* symmetry plane, as measured by the pressure taps that populate the surface of the model.

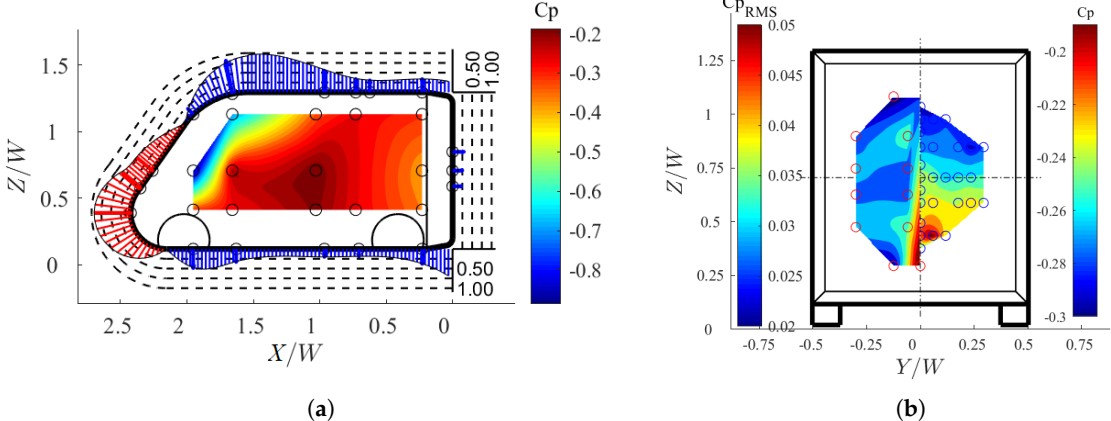

**(a)**           **(b)**

**Figure 4.** Natural flow. (**a**) Mean pressure distribution on the lateral surface and in the symmetry plane (vectors are obtained by interpolating the pressure taps readings with a spline curve), where black circles indicate the physical location of the static probes on the top and lateral surface, respectively. (**b**) Static pressure (right) and fluctuating pressure (left) distribution on the rear face. The location of the actual measurement points is schematically represented using blue and red circles for the static pressure and fluctuating pressure, respectively. The contour representation is obtained by interpolating the measurements on a structured grid. It is worth explicitly mentioning that the blue circles that populate the edges of the model are physically located along the plane of symmetry of the model (i.e., $Y/W = 0$).

The pressure coefficient is defined as

$$c_p(i) = \frac{\overline{p}_i - p_\infty}{0.5\rho V_\infty^2} \tag{3}$$

where $p_i$ is the instantaneous pressure signal of the $i$th pressure tap at time $t$. The free stream static pressure $p_\infty$ coincides with the atmospheric pressure and $\rho$ is the air density. The actual measurements were carried out at the locations indicated by the open circles on the model's side and symmetry plane, respectively. The pressure readings in the symmetry planes were interpolated using a spline curve to obtain the vector plot distribution. Red arrows are representative of high pressure regions, while blue ones indicate suction regions.

The pressure coefficient distribution in Figure 4a shows a behavior in line with experiments reported in the literature [25]: a high pressure region corresponding to the front stagnation point, then featuring a progressive reduction in pressure reaching a minimum value on the top part. The air channeled through the gap between the model and the wind tunnel floor exhibits a pressure reduction near the rear exit of the vehicle due to the sudden area growth.

As expected, the contour plot in Figure 4a exhibits a nearly uniform distribution of the pressure coefficient on the lateral surface of the vehicle, with the exception of the region immediately past the vehicle's front, which presents very low values of the pressure coefficient. The elongated shape of this region in the direction of the slanted surface is a well known feature of such bluff bodies, typically referred to as A-pillar vortex [44].

In Figure 4b, we report the pressure coefficient measured with the taps (right) and the root mean square (RMS) of the pressure signal captured with the microphones (left). The RMS of the pressure coefficient $c_{p_{RMS}}$ is calculated as

$$c_{p_{RMS,i}} = \frac{\sqrt{\frac{1}{N-1}\sum_{i=1}^{N}\left(p_i(t)-\overline{p}_i\right)^2}}{0.5\rho V_\infty^2} \tag{4}$$

Since only half of the model's rear face is instrumented (the right part), we only show data concerning half of the base. For the sake of conciseness, the other half of the model rear face shows the results in terms of pressure fluctuations, thus the plot is mirrored with respect to the actual location of the taps.

Static pressure measurements show that the flow separates from the model edges, producing an extended low pressure region with a pressure gradient directed towards the bottom part of the vehicle. We relate this pressure difference to the effects of the underbody flow and to the interaction between the shear layer emerging from this region and the low pressure region behind the car.

The $c_{p_{RMS}}$ distribution reveals the presence of two regions of maxima in proximity of the top and bottom edges of the vehicle, which can be connected to the oscillation of the vortical structures produced in the wake of the model.

### 3.2. Controlled Flow: Single Jet Forcing

We start by investigating the case of single jet actuation. This helps to assess the effects of the single jets on the flow field; furthermore, it may give an indication on the most effective one in terms of drag reduction. We consider the following actuations: lower jet only, upper jet only, and lateral jets only (which are connected to the same flow meter and then work simultaneously). Figure 5 shows the effect of the velocity ratio $V_j/V_\infty$ on the drag reduction ($DR\% = (C_D - C_D^C)/C_D \cdot 100$, with $C_D^C$ being the drag coefficient in the controlled configuration and $C_D$ the drag coefficient in the natural configuration) when the lower, upper, and lateral jets are alternatively activated.

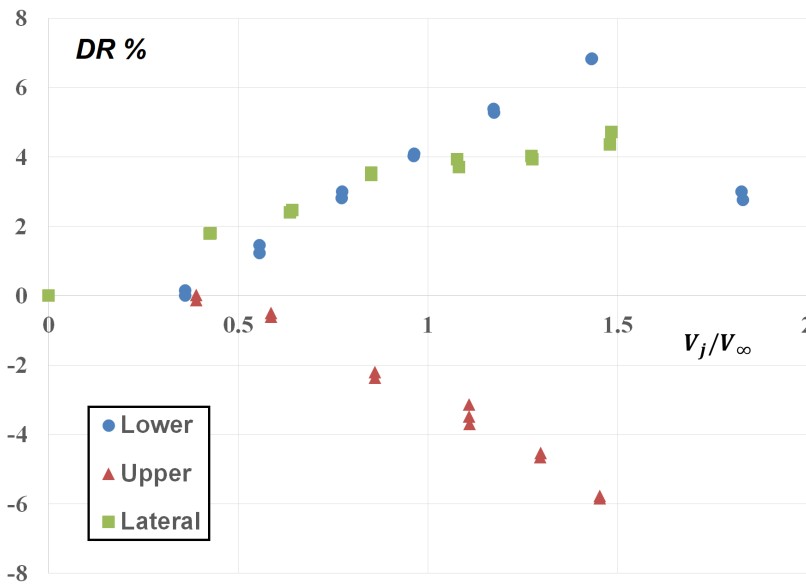

**Figure 5.** Effect of single jets actuation on drag reduction.

Values of the jet velocity ratio $V_j/V_\infty$ smaller than 0.4 are ineffective in terms of drag reduction, regardless if it is the lower, upper, or lateral jet being considered. As $V_j/V_\infty$ exceeds 0.45, increasingly higher drag reductions can be detected, at least for the lower and lateral jet cases. The upper jet activation seems to be detrimental. A recent investigation by Zhang et al. (2018) [16] showed that the blowing angle covers a fundamental role in defining the effectiveness of the flow control system and, more particularly of the top edge jet. In the present case, the choice of the blowing angle was set in a symmetric configuration and the optimization of this parameter is beyond the scope of the paper.

Nevertheless, it is important to point out that further drag reductions can be obtained by optimizing this parameter.

Figure 5 suggests that the most effective forcing is due to the lower jet. Its beneficial effect grows almost linearly, reaching a maximum drag reduction of about 7% for $V_j/V_\infty = 1.4$. For even higher forcing strengths, the effectiveness of the lower jet decreases, evidencing a drag reduction of about 3% for $V_j/V_\infty = 1.8$. The highest efficiency of the lower jet is in line with the fact that the lower part of the wake presents a strong interaction with the flow that undergoes the acceleration through the clearance between the model and the wind tunnel floor. Tampering with this interaction might be a key to achieve strong drag reductions.

The lateral jets, acting as a pair, are also effective in reducing the model's drag, featuring a maximum drag reduction of about 4.7%. In addition, for smaller actuation strengths, the lateral jets are even more effective than the lower one, featuring a DR = 1.8% when $V_j/V_\infty = 0.42$.

In Figure 5, we identify the most effective configurations as: lower jet only $V_j/V_\infty = 1.4$ and lateral jets $V_j/V_\infty = 1.5$. For the sake of brevity, we only report the pressure coefficient ($c_p$) distribution and the RMS ($c_{p_{RMS}}$) of the pressure coefficient on the rear face for these two particular cases in Figure 6a,b.

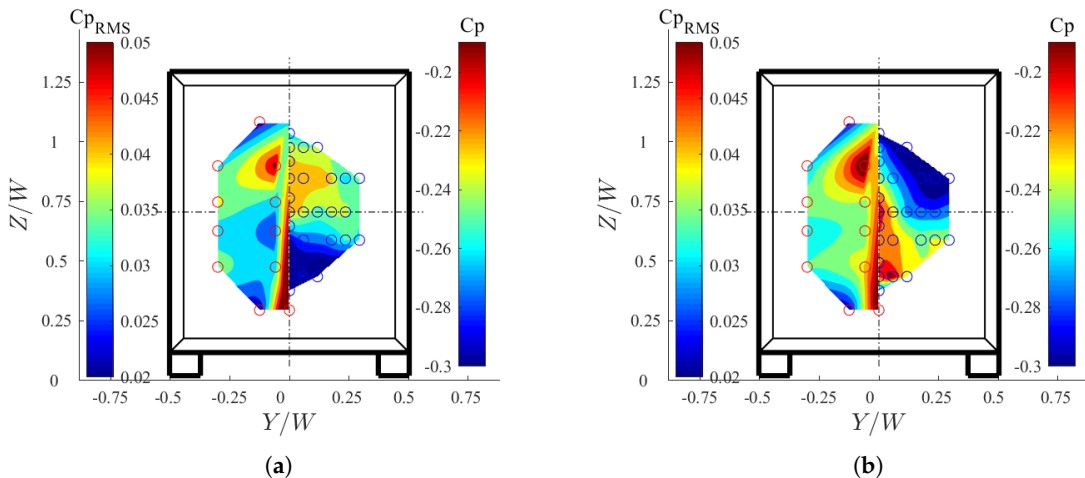

**Figure 6.** Distributions of static pressure coefficient ($c_p$, right) and RMS of the pressure coefficient ($c_{p_{RMS}}$, left) on the rear face: (**a**) lower jet forcing, $V_j/V_\infty = 1.4$; and (**b**) lateral jets forcing, $V_j/V_\infty = 1.5$. The location of the actual measurement points is schematically represented using blue and red circles for the static pressure and fluctuating pressure, respectively. The contour representation is obtained by interpolating the measurements on a structured grid.

Focusing on the case with only lower jets activated, a comparison between Figures 6a and 4b immediately reveals the significantly different $c_p$ distribution in the controlled case. The high-speed fluid injected in proximity of the bottom edge of the vehicle generates a low-pressure region which replaces the high pressure one present in the natural case, but also increases the $c_p$ in the rest of the distribution. As further evidence of the effectiveness of this control, Figure 6a shows that the jets partly suppress the pressure fluctuations across the rear of the model, particularly near the region of injection.

Figure 6b is representative of the case of lateral jets forcing. The high-pressure region near the bottom edge of the vehicle is still present, with little change from the natural flow condition. This control configuration mostly affects the region near the lateral edges. However, as is well evidenced in Figure 6b, significant pressure fluctuations can be detected in this same region, possibly associated with the interaction of the shedding phenomenon (ascribable to the Karman vortex street) with the blowing jets. Even in the controlled cases, as evidenced in Figure 6, the static and RMS pressure distribution show an asymmetric distribution with respect to the *XY* plane [45]. Nevertheless, in the case of lower jet blowing, the pressure gradient reverses its orientation (Figure 6a).

The implications of this different orientation will be further investigated when describing the PIV results.

### 3.3. Controlled Flow: Combined Jets

We now turn our attention to configurations where the jets are actuated simultaneously, with the aim of determining the most efficient and energetically favorable condition. Despite the not encouraging results of the upper jet, we initially kept it as a configuration option due to a possible beneficial interaction with the other jets.

The first configuration we investigated was the symmetric case, i.e., same blowing speeds for all jets ($V_{lat}/V_\infty = V_{low}/V_\infty = V_{up}/V_\infty = V_j/V_\infty$). In Figure 7, we present the measured $DR\%$ as a function of the velocity ratio $V_j/V_\infty$. As the forcing increases the drag reduction also increases up to a velocity ratio $V_{low}/V_\infty = 0.64$ for which the $DR\%$ attains a value of 4.3%. Beyond this point, the control effectiveness reduces almost linearly. Comparing the results of the symmetric forcing to the single jet case, one may conclude that the single jet actuation brings higher benefits, even without accounting for the overall system efficiency.

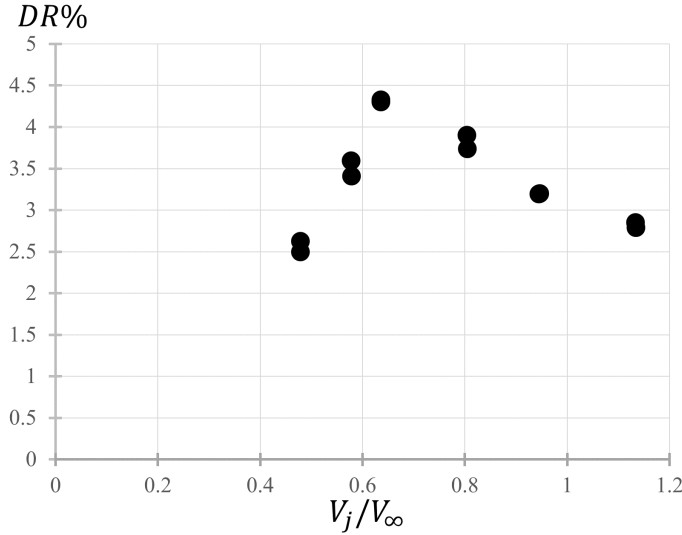

**Figure 7.** Drag reduction for symmetric forcing ($V_{lat}/V_\infty = V_{low}/V_\infty = V_{up}/V_\infty$).

In the light of the results of Figures 5 and 7, we focused on the most efficient control configurations. Only asymmetrical blowing configurations are considered in the remainder of the paper, whereby asymmetrical blowing we mean the employment of only lateral and bottom jets not necessarily at the same injection speed.

We performed a parametric study to determine the most favorable configuration, accounting only for lateral ($V_{lat}/V_\infty$) and lower jets ($V_{low}/V_\infty$). It is worth recalling that the lateral jets, in the present experiment, are always activated simultaneously and with the same injection speed. To cover a significant portion of the parametric space, we varied the jets' speed in the range $0 \leq V_j/V_\infty \leq 1.5$. As evidenced in Figure 8a, high actuation speeds are the most effective in terms of drag reduction as already evidenced in the literature for different vehicle geometries [24]. It is interesting to notice that, similar to what was already proposed when investigating the single jet actuation, the lower jet is still very effective even with little influence of the lateral jets. However, Figure 8a clearly suggests that the combined effect of lateral and lower jets leads to a drag reduction twice larger than the most favorable standalone configurations. The vehicle wake is indeed highly three-dimensional; hence, it is reasonable to expect higher efficiency when acting on the three edges simultaneously, rather than focusing only on the lower part of the wake.

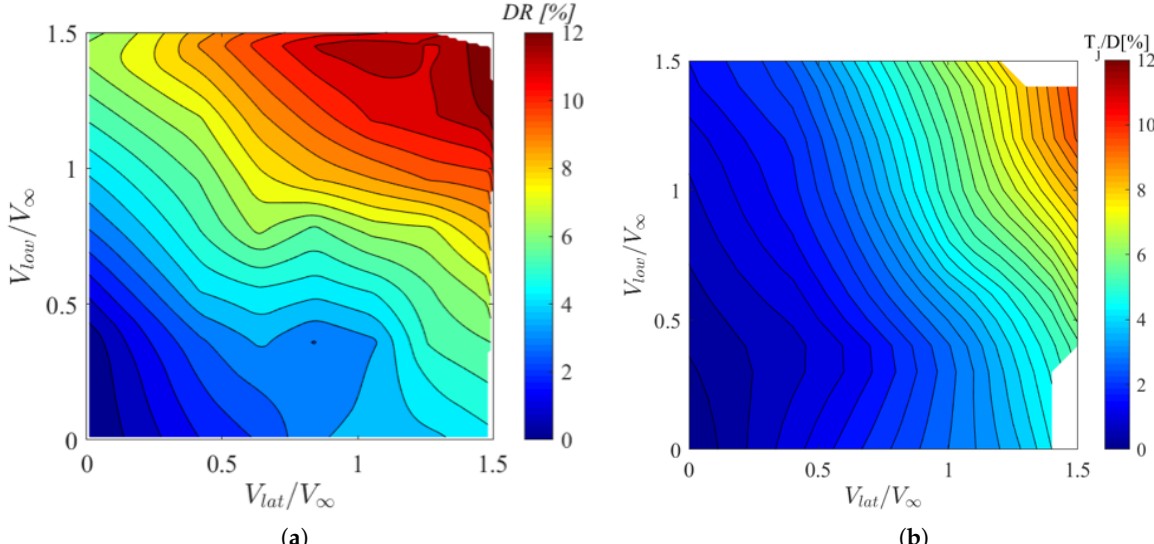

**Figure 8.** (**a**) Map of drag reduction for asymmetrical forcing. Lower and lateral jet effects. (**b**) Effect of the thrust produced by the blowing jets with respect to the drag of the natural case.

It is worth explicitly mentioning that the blowing jets have a secondary beneficial effect on drag reduction, given by the thrust that they exert. In Figure 8b, we plot the contour representation of the thrust produced by the lower and lateral jets normalized with respect to the drag in the natural case. By comparing Figure 8a,b, it is immediate to understand how a large share of drag reduction must be addressed to the effect of the blowing jets at the highest injection speeds. However, their effect is mainly limited to the range of injection speeds $V_{lat}/V_\infty$ >1.3 and $V_{low}/V_\infty$ >1, where $T/D$ exceeds 6%. In Figure 8a ,we identify the configuration leading to the maximum drag reduction (DR% = 12.7%) as $V_{lat}/V_\infty$ = 1.4 and $V_{low}/V_\infty$ = 1.2.

In Figure 9, we report the distribution of the pressure coefficient ($c_p$, right) and of the RMS of the pressure coefficient ($c_{p_{RMS}}$, left) on the rear face of the model for the maximum drag reduction configuration.

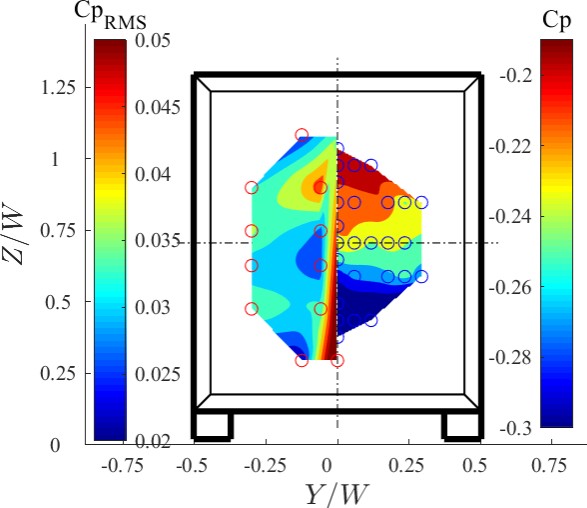

**Figure 9.** Static pressure (right) and RMS pressure (left) distribution on the rear face for maximum *DR*% forcing, $V_{lat}/V_\infty$ = 1.4 and $V_{low}/V_\infty$ = 1.2. The location of the actual measurement points is schematically represented using blue and red circles for the static pressure and fluctuating pressure, respectively. The contour representation is obtained by interpolating the measurements on a structured grid.

The distribution of the pressure coefficient highlights a large region of pressure recovery in the central and upper part of the rear face. This control configuration leads to a qualitatively similar distribution to the one obtained with the only lower jet activated (Figure 6a), thus further confirming that the lower part of the wake plays a dominant role in the total drag budget. Furthermore, also in this case the pressure gradient is directed upwards, differently from the natural case. This suggests that beyond a particular value of the lower jet strength, the flow in the recirculation bubble behind the model is eventually reverted. The $c_{p_{RMS}}$ contour representation in Figure 9 closely resembles the one for the lateral forcing; however, the lower jet activation leads to a significantly smaller activity near the bottom edge. Overall, one may argue that this distribution is a sum of the effects of the two individual distributions obtained by the single activation of the lower and lateral jets as discussed in the previous subsection.

### 3.4. Energy Budget Estimation

Flow control applications in real problems rely on the fact that the proposed solution is efficient in terms of energy budget. Hence, besides the drag reduction achieved in the different configurations, the amount of energy spent must be accounted for.

In this set of experiments, the energy budget was estimated evaluating the energy spent for the flow actuation in terms of losses as well as the corresponding benefit in terms of drag reduction. We introduced the parameter $\zeta$ for the global characterization of the efficiency of the system, defined as the ratio between the power saved $P_s$ and the power consumed $P_c$ for the jets actuation.

$$\zeta = P_s/P_c \tag{5}$$

where we define $P_s = (D - D^c) \cdot V_\infty$ and $P_c = \sum p_{losses} \cdot Q + \sum_{i=1}^{N_{jets}} 0.5\rho V_{j,i}^3 A_i$ where $Q$ indicates the total volumetric flow rate, $A_i$ indicates the cross-section area of the $i$th jet and $V_{j,i}$ the blowing speed of the $i$th jet. Equation (5) is obtained by extending the definition given in many previous investigations [16,24,46] to account also for the losses in the pneumatic line ($\sum p_{losses} \cdot Q$). We do this by estimating the losses in the pipings enclosed within the model using empirical relations [47]. Even though this approach will lead to smaller values of $\zeta$, it will also give an idea of the energy spent in feeding the jets. It is also important to point out that, as also suggested by Choi et al. (2008) [46], a unique definition of the efficiency of a control system cannot be trivially obtained. The definition proposed here is based on the idea of accounting for all the possible losses that are encountered in a realistic implementation of such control mechanism.

An efficient flow control configuration will be characterized by values of $\zeta > 1$. The sensitivity of this performance parameter is strictly related to some features of the wake receptivity with respect to forcing and to the pressure losses. Care was taken to minimize the pressure losses within the model, considering the physical constraints such as the available internal volume. The pressure drops of each component of the external pipe lines were estimated considering the manometers pressure measurements and empirical relations reported by Idel'chik (1987) [47] for all the controlled configurations.

The results in the case of asymmetric blowing are shown in Figure 10 in terms of color map of the efficiency $\zeta$ as a function of the velocity ratio of the lateral and lower jets. The range of actuation velocities $V_{lat}/V_\infty < 0.5$ and $0.5 < V_{low}/V_\infty < 1.5$ is characterized by efficiency values well above unity. As can be observed, the effect of the lateral jets for $V_{lat}/V_\infty > 0.5$ greatly deteriorates the efficiency for any forcing strength of the lower jet. For $V_{lat}/V_\infty < 0.5$, forcing with the lower jet significantly modifies the efficiency, evidencing a rather wide region of high $\zeta$. Nevertheless, it must be pointed out that values of $V_{lat}/V_\infty < 0.25$ require the simultaneous activation of the lower jet with $V_{low}/V_\infty > 0.5$ to be effective.

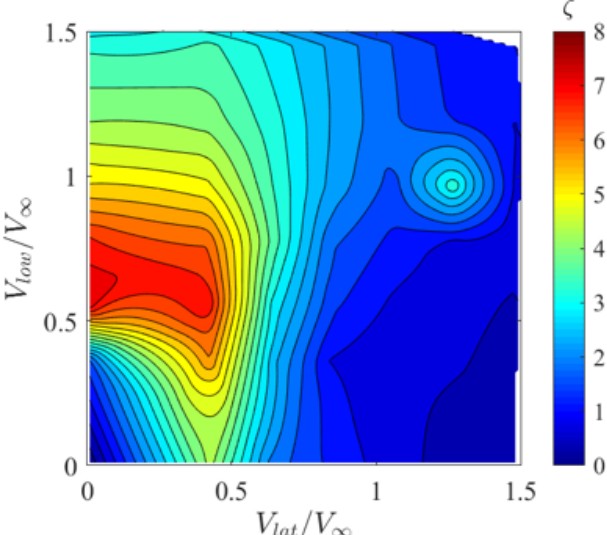

**Figure 10.** Efficiency $\zeta$ of the flow actuation system as a function of the blowing speed of the lower ($V_{low}/V_\infty$) and lateral ($V_{lat}/V_\infty$) jets.

Similar conclusions can be drawn by following the approach proposed by Englar (2001) [48], who investigated the wake sensitivity to the control defining a ratio between the drag variation and the jet momentum coefficient $C_\mu$. It is possible to define a receptivity coefficient

$$\frac{\Delta C_D}{C_\mu} = \frac{D - D^C}{0.5 \sum_{i=1}^{4} \rho A_i V_{j,i}^2} \tag{6}$$

Equation (6) evidences that $\Delta C_D / C_\mu$ represents the ratio between the change in drag force and the sum of the momentum injection of the jets. This ratio can assume positive or negative values according to the sign of the numerator. Positive values mean effective drag reductions ($D^C < D$). Values of $\Delta C_D / C_\mu > 2$ correspond to flow control configurations particularly effective in modifying the wake structure or in others words the wake shows high receptivity with respect to these forcings. On the other hand, this parameter does not deliver any information about the losses related to the control mechanism.

In Figure 11, we report the values of $\Delta C_D / C_\mu$ as a function of the $DR\%$ using the jets in standalone configuration (lower, upper, and lateral).

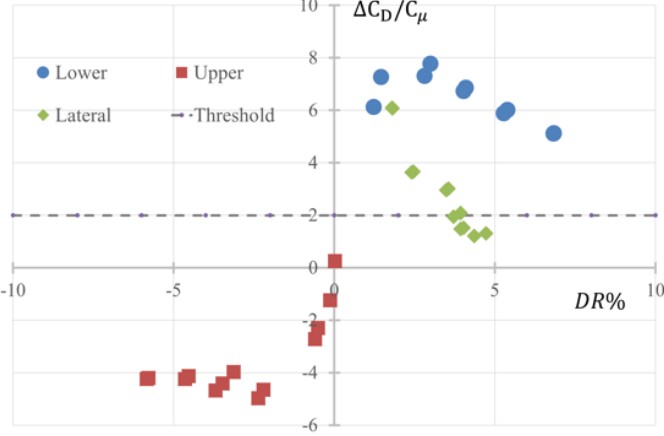

**Figure 11.** Wake receptivity for the single jet forcing configuration.

As can be observed, the lower forcing appears to be the most effective because it favorably modifies the dynamics of the wake–jet interaction for which $\Delta C_D / C_\mu$ attains value close to 7.5 in correspondence of $DR\% \approx 2\%$. Increasing the forcing strength, the wake reduces its favorable reaction with respect to the forcing. Conversely, the results of the lateral jet evidence a poor wake receptivity performance, especially for $DR\% > 3.5\%$ where $\Delta C_D / C_\mu < 2$. The action of the upper jet results to be highly ineffective, for the reasons discussed above.

The same analysis can also be carried out for combined configurations. In Figure 12a, we report the contour representation of the wake receptivity $\Delta C_D / C_\mu$ to the flow control system for asymmetrical forcing as a function of the velocity ratios of the lateral and lower jets. It is immediately possible to verify that a large portion of the investigated configurations leads to values of $\Delta C_D / C_\mu > 2$. In particular, the range of actuation velocities $0.38 < V_{lat}/V_\infty < 0.42$ and $0.1 < V_{low}/V_\infty < 1.4$ corresponds to values of $\Delta C_D / C_\mu > 6$, with a maximum of 8 for $V_{low}/V_\infty \approx 0.6$. The corresponding value of efficiency for this configuration is $\zeta \approx 6$, which gives rise to a drag reduction of about 4%.

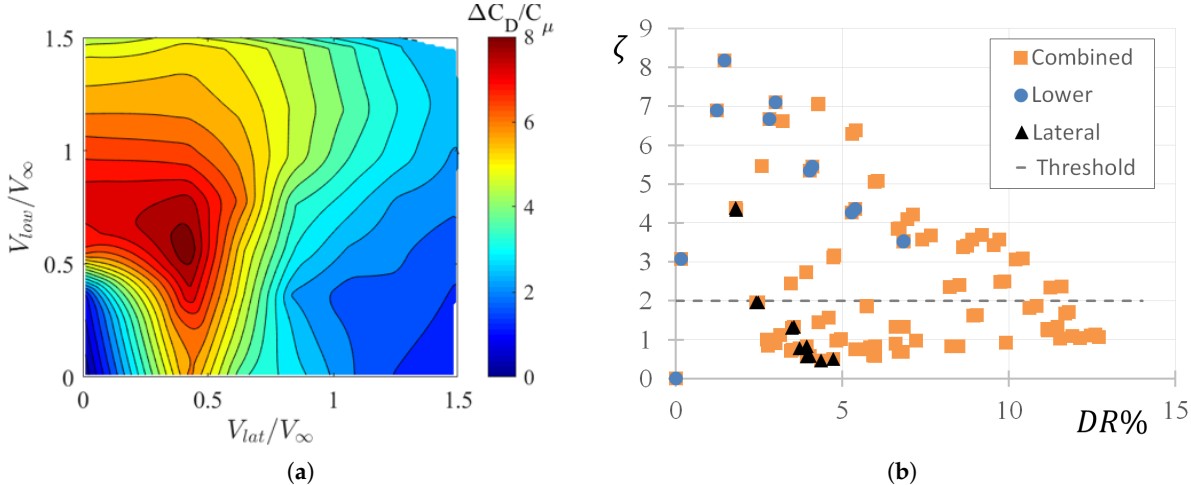

**Figure 12.** Wake receptivity using asymmetrical forcing: (**a**) lower and lateral jets; and (**b**) efficiency vs. $DR\%$ for standalone and combined jets forcing.

Although this value does not correspond to the maximum drag reduction (Figure 12b), it is the most profitable accounting also for the energy spent to feed the control system. Conversely, the maximum drag reduction configurations are characterized by values of the efficiency even smaller than 1.

Although topologically similar, the two maps reported in Figure 10 and Figure 12a show some interesting differences. In particular, even though the location of the maximum efficiency of the system is obtained for the same combination of lateral and lower jets actuation speeds, the wake receptivity seems to suggest that the system features values of wake receptivity larger than 2 almost everywhere. Figure 10 shows instead that as the actuation speed increases, the losses within the pipings (which are proportional to $V_j^2$) may become predominant, hence leading to very small values of $\zeta$.

In Figure 12b, we summarize the values of $DR\%$ and $\zeta$ for the investigated forcing configurations (standalone and combined). From the application point of view, this plot can give an immediate grasp of the drag reduction and of the system efficiency for a given configuration. The corresponding blowing speeds can be obtained looking at Figures 5 and 10 for the standalone or combined configuration, respectively. The data points relative to the lower jet in standalone configurations are characterized by high values of efficiency and satisfactory levels of drag reduction. This is important from an application point of view since the realization costs based on this approach is reduced compared to the combined configuration. For example, a flow control system could be based on the lower jet only, if

drag reduction of the order of 5% are considered as acceptable. On the contrary, the lateral forcing alone appears not convenient at all, as it features low values of drag reduction and of system efficiency.

Table 3 summarizes the most relevant configurations in the case of asymmetrical forcing. We identify two conditions: the best compromise, representing the maximum receptivity and incidentally the value of maximum efficiency $\zeta$, and the maximum drag reduction. We decided to keep this second configuration with the purpose of investigating the flow field in the near wake when the jets alter the most its structure and dynamics, even if the energy budget is not favorable.

**Table 3.** Relevant forcing configurations identified for the current investigation.

|  | **Best Compromise** | **Maximum Drag Reduction** |
|---|---|---|
| $V_j/V_\infty$ | $V_{lat}/V_\infty = 0.4$ and $V_{lat}/V_\infty = 0.6$ | $V_{lat}/V_\infty = 1.4$ and $V_{lat}/V_\infty = 1.2$ |
| $DR\%$ | $\approx 4.5\%$ | $\approx 12\%$ |
| $\zeta$ | 8 | $<1$ |
| $\Delta C_D/C_\mu$ | 8 | 1.2 |

### 3.5. Flow Topology in the Near Wake

The sPIV measurements performed at seven streamwise distances ranging between $0.82W$ and $1.35W$ allow for the three-dimensional reconstruction of the average flow field, as presented in Figure 13; $\omega_i$ indicates the vorticity component along the $i$ axis.

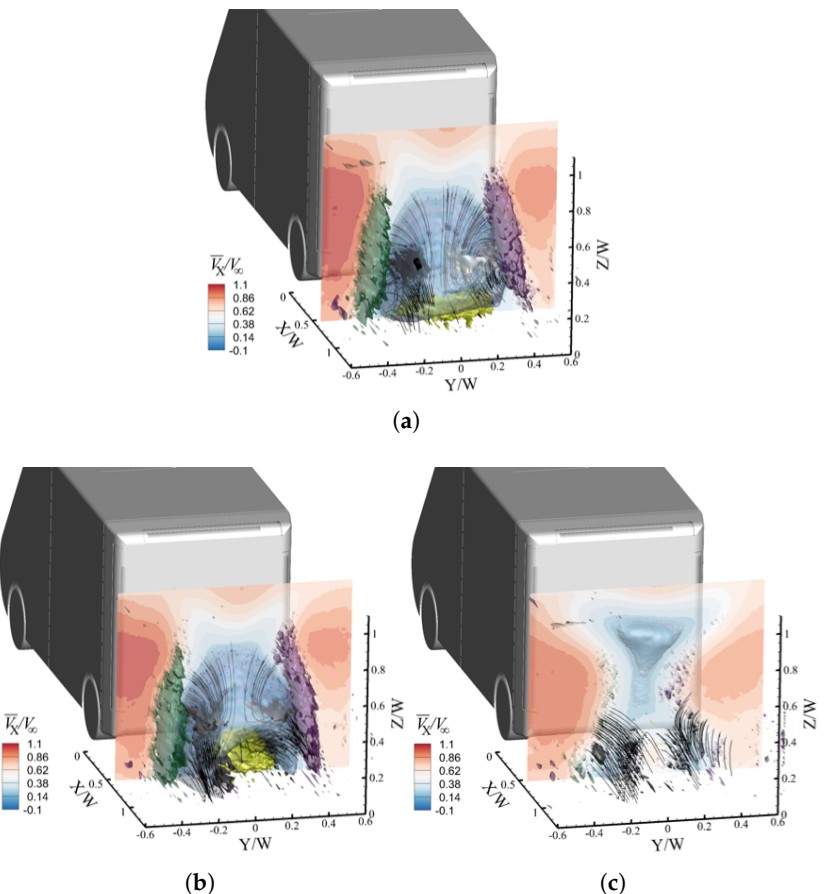

**Figure 13.** Three-dimensional organization of the wake in the three investigated cases: (**a**) natural; (**b**) best compromise; and (**c**) maximum drag reduction. Contour representation of the mean streamwise component $V_x/V_\infty$ at $X/W = 0.82$. Iso-surfaces of $V_x/V_\infty = 0.1$ (blue), $\omega_Z W/V_\infty = \pm 3.2$ (green and purple, respectively); $\omega_Y W/V_\infty = -1.9$ (yellow iso-surface); Iso-surface of Q-criterion $> 0$, color-coded according to values of streamwise vorticity $\omega_X W/V_\infty$.

In the natural case (Figure 13a), the iso-surface $V_x/V_\infty = -0.1$ identifies a well-defined region of low momentum extending up to $X/W = 1.2$, corresponding to the recirculation bubble. The high momentum flow coming from the sides of the model (both lateral and top), when interacting with the low momentum flow in the recirculation bubble, produces regions of high shear. Given the limited extent of the measurement region, only the two shear layers developing from the lateral sides of the model can be clearly measured and identified with the iso-surfaces of $\omega_Z W/V_\infty = \pm 3.2$ (green and purple, respectively). The shear layers cause the entrainment of high-momentum fluid within the recirculation bubble.

As evidenced by the streamlines overlaid to the volumetric measurement, when entrained into the low momentum region, the high momentum flow rolls-up into two counter-rotating streamwise structures, well-evidenced by the iso-surface of positive values of Q (Q-criterion [49]), color-coded according to values of streamwise vorticity $\omega_X W/V_\infty$.

Furthermore, the interplay between the underbody flow, which is characterized by streamwise velocity values larger than the free stream ones, and the low-momentum region, produces a steady vortical structure located at X/W = 0.85, identified by the yellow iso-surface of $\omega_Y W/V_\infty = -1.9$.

As pointed out in the Introduction, tampering with the main structures that characterize the near wake of bluff bodies is the key to achieving high values of drag reduction. In the maximum drag reduction case, the effect of the jets is disruptive in the sense that the flow structures are significantly affected by the momentum injected through the rear base periphery. Further insights about the reorganization of the flow structures can be obtained looking at the pPIV measurements.

The pPIV measurements carried out in the *XZ* plane (Figure 14a,b) show that, in the natural case, a large steady vortex is produced in correspondence of the top edge of the model rear face and characterized by vorticity of opposite sign with respect to the smaller structure, already identified in the three-dimensional measurements. As the underbody flow is characterized by higher momentum than the free stream, the recirculation bubble is not symmetric about the *XY* plane, with a more elongated extent in the streamwise direction near the bottom portion of the vehicle.

Now, we turn our attention to the controlled cases, namely best compromise and maximum drag reduction. The injection speed of the lower and lateral jets are reported in Table 3.

Considering the best compromise case (Figure 13b), a first significant difference can be detected by looking at the iso-surface of streamwise velocity. The effect of the lower jet is such that the bubble is deflated in its lower part. Indeed, the momentum injected from the lower jet reduces the shear rate between the underbody flow and the low momentum region. Conversely, the lateral jets do not significantly affect the flow structure, at least from the qualitative standpoint, thus suggesting that the lower jet is the most effective in terms of drag reduction. Despite this difference, the entrainment of high momentum fluid leads to the roll-up and the production of the two streamwise structures.

The pPIV measurements in the *XZ* plane (Figure 14c,d) reveal that the jet injection in this configuration leads to a reduction of the streamwise extent of the recirculation bubble. In fact, it reduces by about 20% with respect to the natural case.

The maximum drag reduction configuration shows a significantly different structure of the flow field. The jets injection leads to a strong reduction of the lateral shear layers intensities (Figure 13c). The recirculation bubble assumes a double symmetric configuration, with only a small portion of the spanwise extent being interested by low momentum.

Differently from the two previous cases, the effect of the lateral jets is such that the entrained fluid does not roll up, hence avoiding the production of the two counter-rotating streamwise vortices. The pPIV measurements (Figure 14e,f) reveal that the effect of the jets injection significantly reduces the streamwise extent of the recirculation bubble. Furthermore, there is a switch in the mean rotation direction within the recirculation bubble, with the larger vortex being located in the bottom part of the wake; this result is consistent with the base pressure distribution reported in Figure 4b, as well as with the observations of Grandemange et al. (2013) [45].

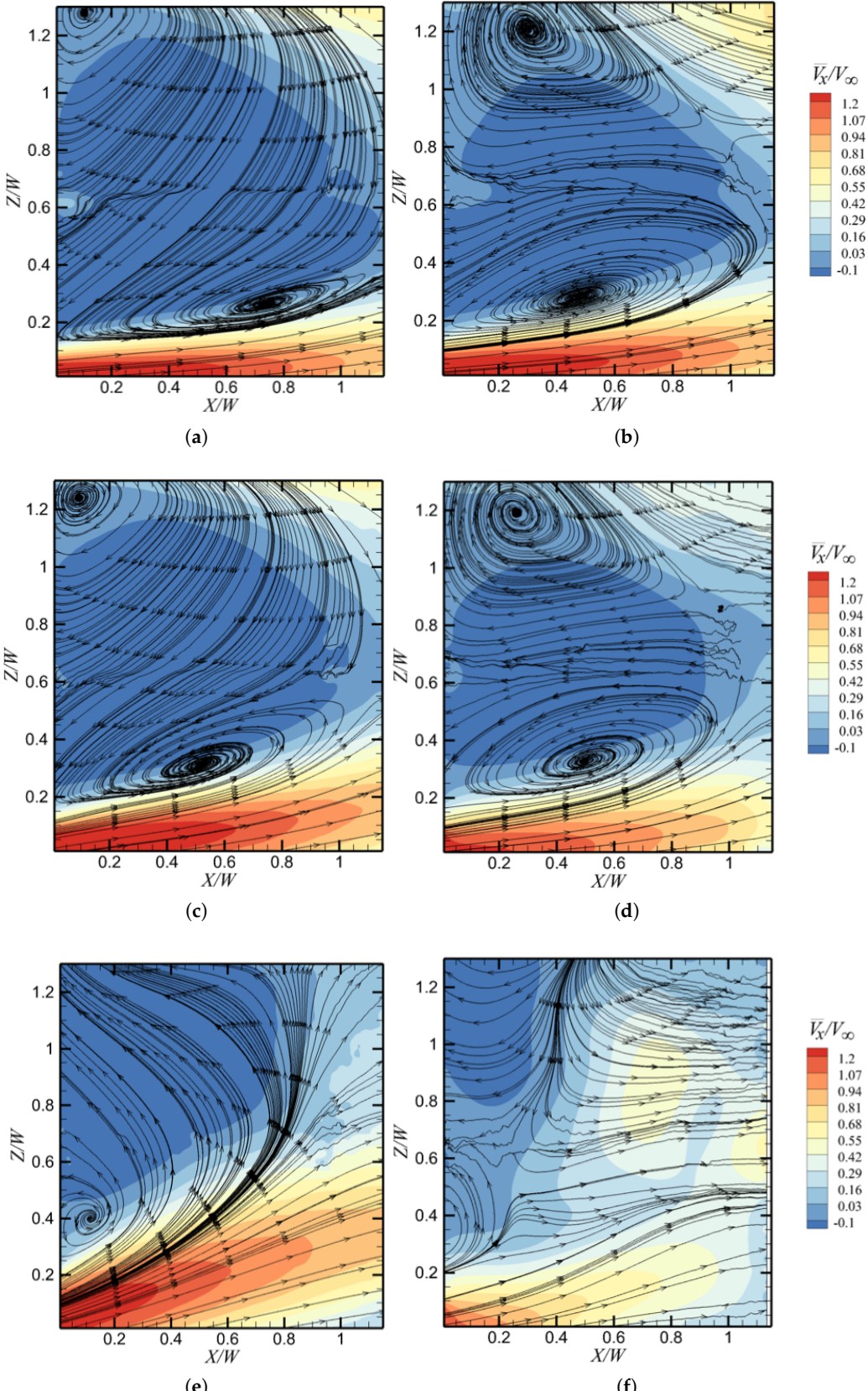

**Figure 14.** Contour representation of the mean streamwise velocity component $V_X/V_\infty$ in the following cases: (**a**,**b**) natural; (**c**,**d**) best compromise; and (**e**,**f**) maximum drag reduction ($Y/W = 0$, left column and $Y/W = 0.2$ right column).

In Figure 15, we report the vertical component of the mean squared velocity fluctuation $\overline{v_z v_z}$ in the natural, best compromise, and maximum drag reduction cases. We present the profiles at a fixed vertical location $Z/W = 0.7$ and at three streamwise locations $X/W = 0.82$ (Figure 15a), $X/W = 1$ (Figure 15b), and $X/W = 1.35$ (Figure 15c).

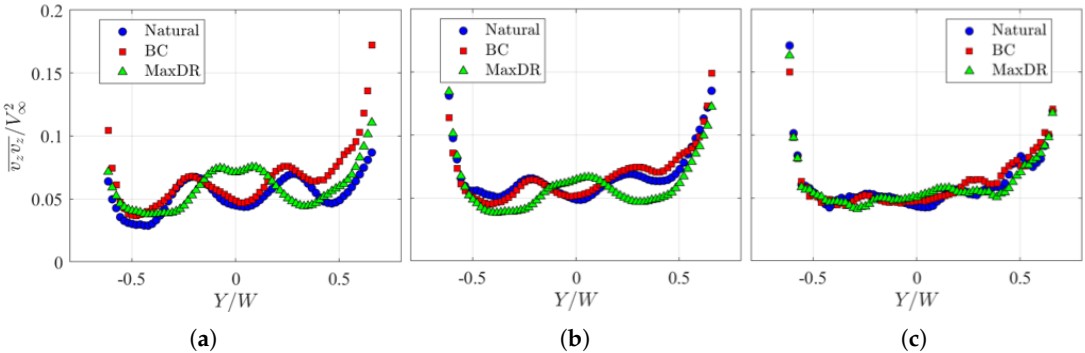

**Figure 15.** Vertical component of the mean squared velocity fluctuation $\overline{v_z v_z}$ in the natural, best compromise, and maximum drag reduction cases. We present the profiles at a fixed vertical location $Z/W = 0.7$ and at three streamwise locations: (**a**) $X/W = 0.82$; (**b**) $X/W = 1$; and (**c**) $X/W = 1.35$.

At the shortest streamwise distance, both the natural and the best compromise case show a double peak shaped profile, with the local maxima located at $Y/W = \pm 0.25$. Comparing these values with the pPIV results reported in Figure 14a–d, it is possible to infer that the location of the local maxima correspond to the region where, on average, the top and bottom vortices interact. Given the highly unsteady character of the near wake, the interplay between the two large structures leads to significant turbulent fluctuations.

The maximum drag reduction case shows a much less pronounced double peak profile, with the maxima located in near proximity of the model centerline. Furthermore, the regions of maxima evidenced in the natural and best compromise cases are replaced by local minima.

For all cases, the profiles attain their global maxima at $|Y/W| > 0.5$, due to the high intensity fluctuations occurring within the shear layers developing from the lateral sides of the model.

At $X/W = 1$, while the natural and best compromise cases still present turbulent fluctuations that are similar to the ones measured at the shortest streamwise distance, the maximum drag reduction case features more attenuated values. This difference is a consequence of the significantly lower extent of the low momentum region developing in the latter case, compared to the natural and best compromise ones. At $X/W = 1.35$, which is outside the recirculation bubble for all cases, all profiles collapse, only showing minor influence of the aforementioned peaks.

### 3.6. Fluctuating Pressure Analysis

The wake structure is further investigated analyzing the pressure fluctuations signals through spectral and Wavelet analysis.

In Figure 16a, we report the plot of the power spectral density (PSD) for Microphones 13 and 16, sketched in Figure 16b, for the natural, best compromise, and maximum drag reduction configurations. The natural flow case exhibits a peak at $f = 6.7$ Hz, corresponding to a Strouhal number $Sr = fW/V_\infty = 0.13$. As is widely reported in the literature [24–26,31,50], this peak is addressed to the shedding occurring in the wake of a 3D bluff body. It is interesting to notice that the peak associated to Microphone 16 (near the lower edge of the vehicle) is more intense than the one related to the lateral one (i.e., Microphone 13), revealing a higher occurrence of vortex shedding from the top-bottom edges, rather than the lateral ones. In the controlled configurations, this peak progressively fades out, with the lowest intensity consistently achieved in the maximum drag reduction case.

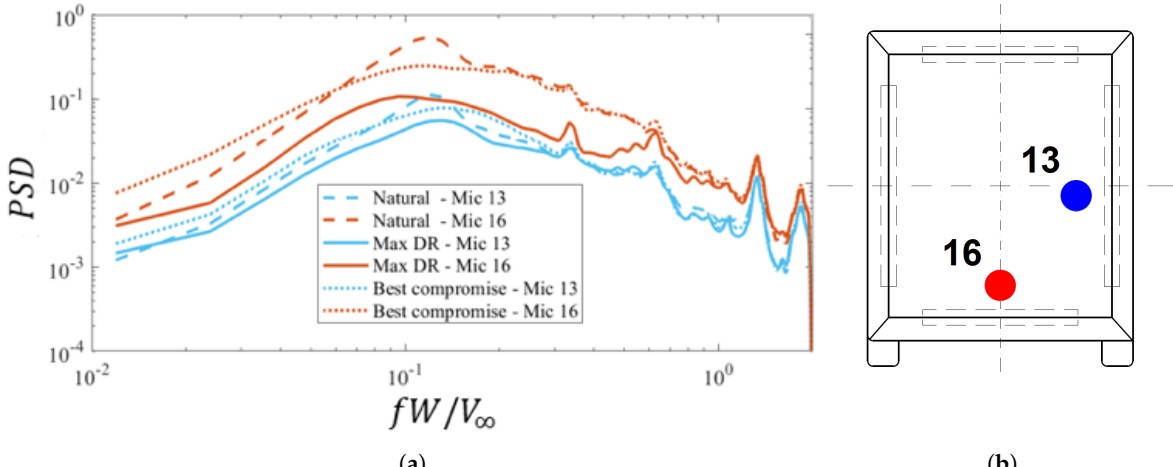

**Figure 16.** (**a**) Power spectral density (*PSD*) of the pressure fluctuations measured by Microphones 13 and 16 in the natural (dashed line), maximum drag reduction (continuous line) and best compromise (dotted line) cases. (**b**) Location of Microphones 13 and 16 on the model's rear base.

The wavelet analysis is a powerful tool and results are valuable due to the possibility of describing the flow behavior in terms of time-scales for the evolution of energy containing structures. Many applications of such a tool applied to fluid dynamics problems are present in the literature. A complete overview of the wavelet analysis is beyond the scope of the following work and can be found in [51].

In Figure 17, we report the results of the application of the wavelet analysis to the pressure signal detected by Microphones 16 (Figure 17a,c,e) and 13 (Figure 17b,d,f), color-coded according to the energy value associated to a particular frequency and time. The results are obtained for three different configurations, namely natural (first row), best compromise (second row), and maximum drag reduction (last row). In the natural case, Figure 17a, it is immediately clear how a large amount of energy is contained within regularly time spaced structures, occurring at a frequency close to 7 Hz ($S_r = 0.13$, evidenced with a dashed red line), that we associate to the shedding phenomenon as also evidenced form the spectra. The energy level associated with these coherent structures shows significant variations that we associate with the different strength of the vortices. This behavior is linked to the three-dimensional unsteady interaction that takes place on this edge of the model. In addition to this organized content, a considerable amount of energy is also attributed to frequencies higher than the shedding one.

When the control mechanism is operated in the maximum drag reduction configuration (Figure 17e), we detect a remarkable reduction of the peak intensities over all the range of frequency scales, with the only presence of non-periodic peaks at frequencies higher than the shedding one. Nevertheless, as already pointed out, the maximum drag reduction configuration is not a good choice from the efficiency standpoint. We then turn our attention to the best compromise case, reported in Figure 17c. As expected, the wavelet analysis still features some of the peaks outlined in Figure 17a, although with very little energy contained in the frequencies close to 7Hz.

A striking difference between the results obtained from the wavelet analysis applied to Microphones 16 and 13 is the significantly lower energy level associated with the latter (Figure 17b,d,f). Despite the lower energy values featured by Microphone 13, the overall physical behavior is similar to the one that characterizes the natural, best compromise, and maximum drag reduction cases obtained from Microphone 16. The blowing jets weaken the shedding as the forcing strength increases and introduce local peaks at higher frequencies, i.e., at smaller scales.

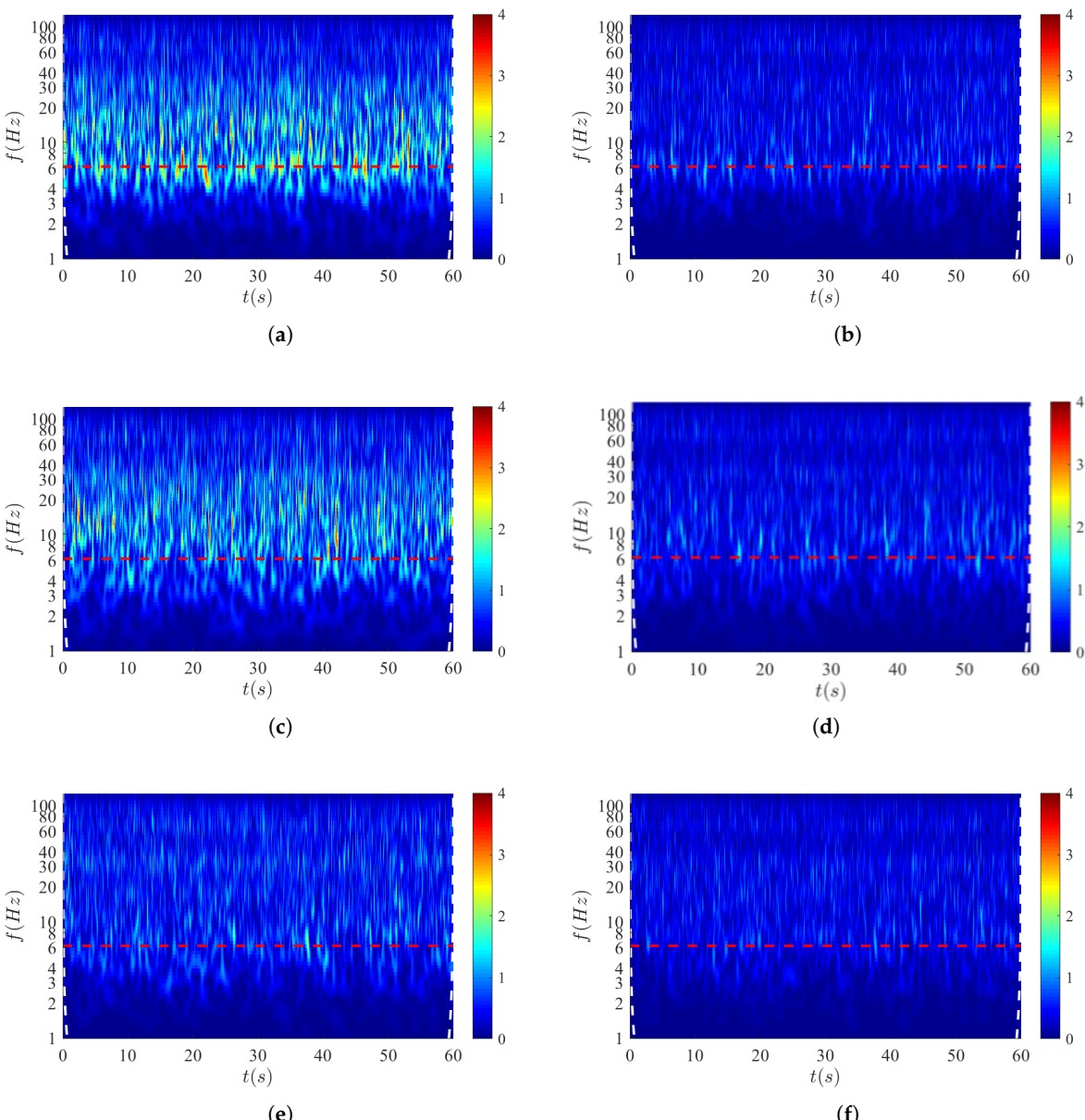

**Figure 17.** Wavelet analysis for signal from Microphone 16: (**a**) natural flow, (**c**) best compromise; and (**e**) maximum DR. For Microphone 13: (**b**) natural flow; (**d**) best compromise; and (**f**) maximum DR. Plots are color-coded according to the magnitude of the wavelet transform of the corresponding pressure signal.

We further investigated the wake dynamics by looking at the cross correlation between the time signals measured by Microphones 13 and 16 , presented in Figure 18. We did this for the three cases of natural flow, maximum drag reduction, and best compromise.

It is interesting to notice that the strong correlation featured by the natural case is progressively lost in the two controlled cases; both cases show a peak at low values of the non-dimensional time $tV_\infty/W$, which we associate with the continuous forcing. However, this peak immediately fades out, with total decorrelation for $tV_\infty/W > 5$. Interestingly, all three cases also feature a very low frequency oscillation, corresponding to a period of the order of 2.5 s, which modulates the shedding signal. This low frequency is of the order of 0.4 Hz and leads to a Strouhal number of $S_r = 0.0076$ that we

attribute to the wake "pumping" [31], an unsteady low frequency phenomenon associated to the global oscillation of the wake [26].

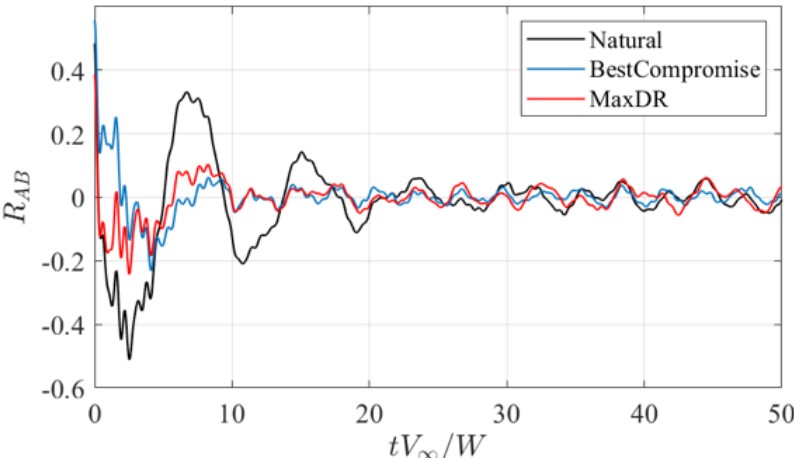

**Figure 18.** Cross correlation of the time signals measured by Microphones 13 and 16 as a function of the normalized time $tV_\infty/W$.

Relevant information about the wake dynamics can also be deduced by the analysis of the taps mounted on the rear base of the vehicle. In particular, we look at the effect of the control mechanism on the stability of the wake. Starting from the pressure signals acquired by the four taps reported in Figure 19b, we can define the pressure gradients across the model base

$$g_Y(t) = \frac{c_{pL}(t) - c_{pR}(t)}{\Delta(Y/W)} \qquad (7)$$

$$g_Z(t) = \frac{c_{pU}(t) - c_{pD}(t)}{\Delta(Z/W)} \qquad (8)$$

Following Bonnavion and Cadot (2018) [27], we can compute the complex base pressure gradient as $g^* = g_Y + ig_Z$, where $i^2 = -1$. Relevant information can be extracted looking at the modulus and the phase of the complex gradient, defined as $g_r = mod(g^*)$ and $\phi = arg(g^*)$, respectively. In particular, the occurrence of a bistable behavior can be inferred from the pdf of the time signals $g_r(t)$ and $\phi(t)$.

Figure 19a shows the pdf of these two quantities for the natural (black line), maximum drag reduction (red line), and best compromise cases (blue line).

First, it is interesting to notice how the mean pressure gradient across the model's rear base attains similar values in the natural and best compromise case, whereas the maximum drag reduction case features a nearly double value of the pressure gradient. From the results in Figure 19, we can draw some conclusions on the effect of the control mechanism on the bi-stable behavior of the wake. Even though the two peaks are not as well demarked as in the case of Bonnavion and Cadot (2018) [27], it can be argued that the natural wake has preference towards two states showing a phase offset equal to $\pi$. We address the less pronounced bi-modal distribution of pdf ($\phi$) with respect to other investigations [25,27,50] to the different geometry of the model's rear base, presenting slant angles on all the four edges.

Furthermore, the combination of the aspect ratio of the rear base and ground clearance is such that the wake is in the interfering region presented in [25], with no clear bi-stable behavior. The best compromise case is instead characterized by a nearly uniform distribution of the pdf ($\phi$), thus suggesting the absence of a well-demarked bi-stable behavior. This aspect is stressed even further in the maximum drag reduction case: a strongly preferential direction of the wake is indeed denoted by the pdf ($\phi$), with only little occurrence probability of different orientations.

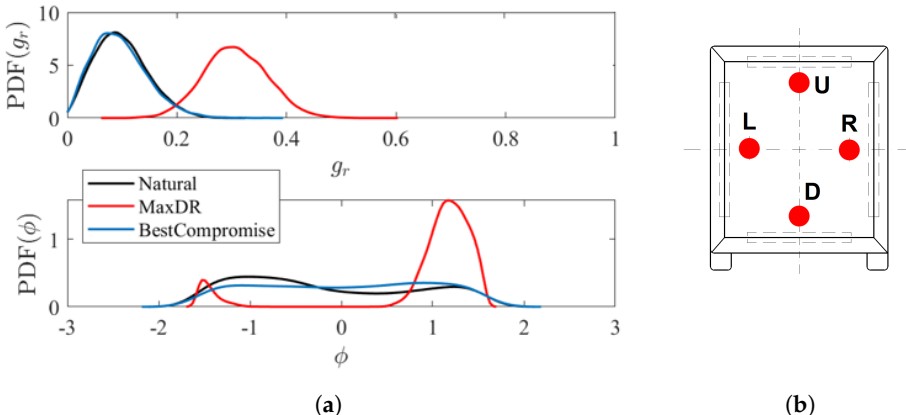

**Figure 19.** (**a**) Probability density function of the modulus and phase of the pressure gradient across the model's rear base. (**b**) Location of the static pressure taps used to calculate the rear base pressure gradients.

## 4. Conclusions

We carried out the experimental investigation on the effects of an active control system on the wake of a square-back vehicle model. For the non-controlled case, the model presents $C_D = 0.465$, in agreement with the typical values for this kind of vehicles.

The standalone configurations, i.e., single continuously blowing jets, showed that the higher benefits are evidenced when only the lower jet is activated, featuring drag reduction values of the order of 4% for $V_j/V_\infty \approx 1$. The asymmetric forcing, obtained combining lower and lateral jets, leads to significantly better results with DR% of about 12.7% for $V_{lat}/V_j = 1.4$ and $V_{low}/V_j = 1.2$. The values of $DR\%$ found in the present investigation are in good agreement with those typical for continuously blowing jets [16,24].

The energy budget analysis highlighted a subset of forcing configurations able to render the wake control profitable from the point of view of potential applications, providing a net benefit to the vehicle. Particularly interesting is the forcing characterized by moderate values of drag reduction (4.5%) but high values of the efficiency $\zeta$ (best compromise, $\approx 8$), thus making this solution particularly amenable for real applications.

Scanning stereo-PIV experiments allowed for the three-dimensional reconstruction of the flow field structure in the near wake. The interplay between low momentum in the wake and high momentum fluid entrained from the sides of the model causes the presence of two counter-rotating streamwise vortices in the natural and best compromise cases. The significant reduction of the shear rate on either sides of the model due to the jet injection suppresses these structures in the maximum drag reduction case. A progressive shortening of the recirculation bubble is detected in the controlled configurations, starting from 1.2 $W$ in the uncontrolled case and reducing down to 0.8 $W$ in the maximum drag reduction one.

An interesting effect evidenced by the best compromise configuration is the capability of achieving relevant drag reductions with minimum disruption of the near wake topology. The stereo-PIV measurements indeed reveal how the major difference stands in the shear rate reduction in the region of interplay between the underbody flow and the low momentum region in the model's wake.

**Author Contributions:** Conceptualization, C.S., J.J.C. and G.I.; methodology, C.S., J.J.C., G.C. and G.I.; software, C.S., J.J.C. and G.C.; validation, C.S., J.J.C. and G.C.; formal analysis, C.S., J.J.C. and G.C.; investigation, C.S., J.J.C. and G.C.; resources, G.I.; data curation, C.S., J.J.C. and G.C.; writing–original draft preparation, C.S., J.J.C., G.C. and G.I.; writing–review and editing, C.S., J.J.C., G.C. and G.I.; visualization, C.S., J.J.C. and G.C.; supervision, G.I.; project administration, G.I.; funding acquisition, G.I. All authors have read and agreed to the published version of the manuscript.

**Funding:** This research received no external funding.

**Acknowledgments:** The authors wish to thank Tommaso Astarita who kindly provided us with the software used to acquire the PIV images as well as the one to perform the cross-correlation analysis. The authors also wish to thank the laboratory technicians Ing. M. Cannata and M. Grivet for the invaluable help they provided with the experimental setup.

**Conflicts of Interest:** The authors declare no conflict of interest.

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
