# Peer review of "Active Flow Control on a Square-Back Road Vehicle"

_fluids, doi:10.3390/fluids5020055_

Round 1

Reviewer 1 Report

I made an effort to make suggestions about grammar and usage throughout, but probably didn't address everything.  Editing would improve readability.  I am concerned about the fact that the floor of the tunnel is stationary (no moving road) considering some of your observations.  I am also concerned about the strut mounting for the same reason.  Some effort to explain why this isn't important or how you have convinced yourselves that the results are valid even with those issues is necessary.  Overall, your approach to ground the study in the practical efficiency of the system is great.  Very necessary for the advancement of active flow control.  Your use of advanced experimental methods is also quite nice.

Author Response

Dear Reviewer,

The response to your review is on the Reviewer1.pdf file.

Best regards.

Author Response

Dear Reviewer,

The response to your review is on the Reviewer2.pdf file.

Best regards.

Round 2

Reviewer 1 Report

No further comments.

Reviewer 2 Report

The authors have addressed most of my comments and I am happy to recommend the paper for publication.